# OncoTherad^®^ (MRB-CFI-1) Nanoimmunotherapy: A Promising Strategy to Treat Bacillus Calmette–Guérin-Unresponsive Non-Muscle-Invasive Bladder Cancer: Crosstalk among T-Cell CX3CR1, Immune Checkpoints, and the Toll-Like Receptor 4 Signaling Pathway

**DOI:** 10.3390/ijms242417535

**Published:** 2023-12-15

**Authors:** João Carlos Cardoso Alonso, Bianca Ribeiro de Souza, Ianny Brum Reis, Gabriela Cardoso de Arruda Camargo, Gabriela de Oliveira, Maria Izabel de Barros Frazão Salmazo, Juliana Mattoso Gonçalves, José Ronaldo de Castro Roston, Paulo Henrique Ferreira Caria, André da Silva Santos, Leandro Luiz Lopes de Freitas, Athanase Billis, Nelson Durán, Wagner José Fávaro

**Affiliations:** 1Laboratory of Urogenital Carcinogenesis and Immunotherapy (LCURGIN), Universidade Estadual de Campinas (UNICAMP), Campinas 13083-865, São Paulo, Brazil; gcacamargo0@gmail.com (G.C.d.A.C.); gabriela.ufscar@gmail.com (G.d.O.); mabelfrazao@hotmail.com (M.I.d.B.F.S.); jumattosogg@gmail.com (J.M.G.); rroston@uol.com.br (J.R.d.C.R.); phcaria@unicamp.br (P.H.F.C.); andre.s.s@me.com (A.d.S.S.); nelsonduran1942@gmail.com (N.D.); 2Paulínia Municipal Hospital, Paulínia 13140-000, São Paulo, Brazil; 3Obstetrics & Gynecology Department, Ovarian Cancer Research Group University of British Columbia, Vancouver, BC V6Z 2K8, Canada; bianca.ribeiro@ubc.ca; 4Diagnosis and Surgery Department, Dentistry School, São Paulo State University (UNESP), Araraquara 14801-903, São Paulo, Brazil; iannybrumreis@yahoo.com.br; 5Pathology Department, Medical School, Universidade Estadual de Campinas (UNICAMP), Campinas 13083-888, São Paulo, Brazil; leandrollfreitas@gmail.com (L.L.L.d.F.); athanase@unicamp.br (A.B.)

**Keywords:** bladder cancer, immunotherapy, Oncotherad, bacillus Calmette–Guerin, nanotechnology

## Abstract

This study assessed the safety and efficacy of OncoTherad^®^ (MRB-CFI-1) nanoimmunotherapy for non-muscle invasive bladder cancer (NMIBC) patients unresponsive to *Bacillus Calmette-Guérin* (BCG) and explored its mechanisms of action in a bladder cancer microenvironment. A single-arm phase I/II study was conducted with 44 patients with NMIBC who were unresponsive to BCG treatment. Primary outcomes were pathological complete response (pCR) and relapse-free survival (RFS). Secondary outcomes comprised response duration and therapy safety. Patients’ mean age was 65 years; 59.1% of them were refractory, 31.8% relapsed, and 9.1% were intolerant to BCG. Moreover, the pCR rate after 24 months reached 72.7% (95% CI), whereas the mean RFS reached 21.4 months. Mean response duration in the pCR group was 14.3 months. No patient developed muscle-invasive or metastatic disease during treatment. Treatment-related adverse events occurred in 77.3% of patients, mostly grade 1–2 events. OncoTherad^®^ activated the innate immune system through toll-like receptor 4, leading to increased interferon signaling. This activation played a crucial role in activating CX3CR1^+^ CD8 T cells, decreasing immune checkpoint molecules, and reversing immunosuppression in the bladder microenvironment. OncoTherad^®^ has proved to be a safe and effective therapeutic option for patients with BCG-unresponsive NMIBC, besides showing likely advantages in tumor relapse prevention processes.

## 1. Introduction

Bladder cancer is a significant global public health issue with high morbidity and mortality rates [1,2]. It is known for its tendency to recur and progress, even in localized disease stages [2,3]. Patients diagnosed with bladder cancer often require several intravesical treatments, whereas advanced and metastatic cases need complex surgical and systemic therapies. Consequently, bladder cancer poses a substantial burden on healthcare resources and it results in significant economic costs [2,4].

Non-muscle-invasive bladder cancer (NMIBC) accounts for most bladder tumors (70–75%). NMIBC is confined to the urothelium as a papillary tumor (pTa) or carcinoma in situ (pTis or CIS) without stromal invasion or with limited invasion into the lamina propria (pT1) [5,6,7]. This cancer type encompasses a heterogeneous group of tumors whose rates of progression to the muscle-invasive phenotype range from 0.8% to 50% within 5 years [8,9]. According to previous studies, significant risk factors for NMIBC progression comprise the combined presence of CIS, a high tumor grade, and the T1 stage [8,9]. Moreover, multiplicity, tumor size ≥3 cm, and history of relapse are also considered risk factors for this disease [8,9].

The therapeutic approach established to manage high-risk non-muscle-invasive bladder cancer (HR-NMIBC) typically involves initial transurethral resection of the bladder tumor (TURBT), which is followed by intravesical instillations of Bacillus Calmette–Guérin (BCG) [7,10,11]. BCG represents the most efficacious intravesical therapy for individuals diagnosed with HR-NMIBC [7,10,11,12,13,14]. Despite an initial complete response to BCG observed in approximately 80% of patients, more than half of these individuals experience disease recurrence and progression within the initial year of treatment. Consequently, a substantial proportion of patients eventually develop what is referred to as BCG-unresponsive disease [7,11,12]. Disease progression can increase mortality rates and delay the timely administration of appropriate radical interventions. Consequently, this process affects patients’ overall survival [7,13,14]. Hence, the early identification of patients prone to treatment failure before starting BCG therapy is of paramount importance, since it can enable these individuals to significantly benefit from early radical cystectomy, notwithstanding its potential impact on their quality of life [14]. Moreover, it is essential emphasizing the global shortage of BCG, as well as the pressing need of finding alternative therapeutic modalities [11,15].

Radical cystectomy stands as the most definitive treatment for NMIBC in this particular context [13,14,15]. However, this procedure is linked to a significant perioperative morbidity rate, and a considerable number of patients either lack the willingness or the capability to undergo such a comprehensive intervention [13,14,15]. Numerous agents have undergone assessment as potential intravesical therapies following BCG treatment [11,15]. Regrettably, none of these agents has demonstrated a consistently robust and enduring response. Consequently, there remains a pressing unmet clinical need for the development of an intravesical treatment that is both effective and durable, while also being safe. This need is particularly critical for patients who wish to avoid the extensive surgical intervention of radical cystectomy.

The therapeutic approach established to manage high-risk non-muscle-invasive bladder cancer (HR-NMIBC) typically involves initial transurethral resection of the bladder tumor (TURBT), which is followed by intravesical instillations of Bacillus Calmette–Guérin (BCG) [7,10,11]. Although the initial response to BCG treatment is often favorable, its long-term effectiveness is only moderately effective, and it leads to a substantial risk of disease relapse and progression [7,11,12]. Disease progression can increase mortality rates and delay the timely administration of appropriate radical interventions. Consequently, this process affects patients’ overall survival [7,13,14]. Hence, the early identification of patients prone to treatment failure before starting BCG therapy is of paramount importance, since it can enable these individuals to significantly benefit from early radical cystectomy, notwithstanding its potential impact on their quality of life [14]. Moreover, it is essential to emphasize the global shortage of BCG, as well as the pressing need of finding alternative therapeutic modalities [11,15].

Modulating the tumor microenvironment (TME) and enhancing T cell infiltration into tumors are pivotal strategies to assist in managing different tumor types [16]. One of the effective approaches involves using innate immunity modulators, more specifically, toll-like receptor (TLR) agonists [16,17]. These agonists aid in increasing the activity of anti-tumor effector cells, such as cytotoxic T cells and natural killer (NK) cells, while simultaneously suppressing immunosuppressive cell populations, such as regulatory T cells (Tregs) and myeloid suppressor cells. Furthermore, they play a significant role in influencing immune cells’ recruitment within the tumor microenvironment [16,17]. Checkpoint inhibitor therapies’ primarily function is to enhance antitumor responses mediated by CD8^+^ T cells. Chemokine receptor 1, CX3C (CX3CR1), was recently identified as a T cell differentiation marker [18]. Notably, CX3CR1 receptors are consistently expressed in CD8+ T cells throughout their differentiation process from CX3CR1^−^CD8^+^ T cells during the effector phase [18]. This feature turns CX3CR1 into a valuable biomarker, mainly in comparison to molecules presenting transient expression patterns in T cells.

TLRs’ activation triggers a complex signaling cascade that activates both innate and adaptive immune responses and releases pro-inflammatory cytokines [16,17]. Upon activation, TLRs recruit adapter proteins presenting toll/interleukin-1 receptor (TIR) domains. These critical TLR signaling adapters encompass myeloid differentiation factor 88 (MyD88), toll/interleukin-1 receptor (TIR) domain-containing adapter protein (TIRAP), toll/interleukin-1 receptor (TIR) domain-containing adapter-inducing interferon-β (TRIF) and translocating chain-associating membrane protein (TRAM) [16,17,19]. MyD88 works as common adapter for all TLRs, except for TLR3. MyD88 activation triggers nuclear factor-kappa B (NF-κB) and mitogen-activated protein kinase (MAPK) pathways, and ultimately produces inflammatory cytokines [16,17,19,20]. TLR4 recruits TRIF through TRAM, whereas TLRs 2 and 4 use TIRAP to recruit MyD88 [16,17,20].

TLR signaling can be classified into two different pathways, namely, the MyD88-dependent (canonical) pathway, which promotes the production of inflammatory cytokines, such as tumor necrosis factor-alpha (TNF-α), as well as interleukins such as IL-6, IL-1β, and IL-10; and the TRIF-dependent (non-canonical) pathway, which activates interferon regulatory factors (IRF-3, IRF-5, and IRF-7)—this process results in interferon production [16,17,19,20]. More specifically, TLR4 uses all four adapter molecules (MyD88, TRIF, TIRAP, and TRAM) to trigger its signaling cascade [16,17,19,20].

The emergence of novel immunotherapeutic agents has revolutionized the treatment applied to different tumor types, such as urothelial carcinoma [10]. These immunotherapies have shown promising outcomes in bladder cancer cases. Accordingly, our research team has developed the OncoTherad^®^ (MRB-CFI-1) nanoimmunotherapy, which recorded favorable outcomes in cancer therapy, mainly in bladder cancer cases [21,22,23,24,25,26]. OncoTherad^®^ (MRB-CFI-1) comprises nanometric components formed by phosphate and metal salts (CFI-1) linked to glycosidic proteins (P14 and P16 proteins) whose size ranges from 420 nm to 530 nm [21,27]. It holds granted patents both in the United States (USPTO: US-11623869-B2; US-11572284-B2; US-11136242-B2; US-11639294-B2) and in Brazil (INPI: BR102017012768B1) [21,28,29,30,31,32].

OncoTherad^®^ (MRB-CFI-1) triggers the human innate immune system by activating TLR2 and TLR4. This process enhances the IFN signaling pathway activation process, which involves TLR4, TRIF, IRF-3, IFN-α, and IFN-γ [26,27,28,33,34,35,36]. More specifically, OncoTherad^®^ (MRB-CFI-1) induces immune stimulation through TLR2. TLR4 takes place by phosphorylating hydroxylated amino acids such as tyrosine, threonine, and serine using compounds containing phosphate salts. This activation process stimulates the stimulator of interferon genes (STING) and leads to increased IFN-α and IFN-γ production [21,26,27,28,29,30,31,32,35,36]. Moreover, OncoTherad^®^ (MRB-CFI-1) downregulates the expression of both the receptor activator of nuclear factor-κB (RANK) and the receptor activator of the nuclear factor-κB ligand (RANK-L) system. Therefore, it aids in preventing metastasis formation and inhibiting disease progression [24,26,27,28,37].

The aims of the current study were to assess both the safety and the effectiveness of OncoTherad^®^ (MRB-CFI-1) nanoimmunotherapy applied to patients with NMIBC who were unresponsive to BCG, as well as to investigate the underlying action mechanisms of this nanoimmunotherapy in the intricate microenvironment of bladder cancer cases by focusing on interrelated factors and signaling pathways.

## 2. Results

### 2.1. OncoTherad^®^ (MRB-CFI-1) Nanoimmunotherapy Primarily Activated Toll-Like Receptor 4 (TLR4)

OncoTherad^®^ (MRB-CFI-1) samples used at the concentration of 50 µg/mL led to significant expression of both TLR4 and TLR2, which accounted for approximately 84.2% and 52.8% of control ligands, respectively (Figure 1). Furthermore, OncoTherad^®^ (MRB-CFI-1) samples used at the concentration of 5 µg/mL presented a remarkable expression effect on TLR4 and accounted for approximately 49.5% of control ligand responses (Figure 1). On the other hand, OncoTherad^®^ (MRB-CFI-1) samples did not trigger significant TLR5, TLR7, TLR8, or TLR9 expression (Figure 1). However, they had a minor expression effect on TLR3, at both 50 µg/mL and 5 µg/mL concentrations, and it corresponded to approximately 10.7% and 9.3% of control ligands, respectively (Figure 1).

On the other hand, P14-16 protein samples had a significant expression effect on human TLR2 at the concentration of 5 µg/mL and accounted for approximately 32.9% of control ligand responses (Appendix A). Furthermore, P14-16 protein samples led to significant TLR2 and TLR4 expression at the concentration of 50 µg/mL. They accounted for approximately 78.4% and 38.2% of control ligand responses, respectively (Appendix A). However, P14-16 protein did not induce significant human TLR3, TLR5, TLR7, TLR8, or TLR9 expression (Appendix A).

Inorganic component (CFI-1) samples of OncoTherad^®^ (MRB-CFI-1) had a significant expression effect on human TLR4 at the concentration of 5 µg/mL and accounted for approximately 25.0% of control ligand responses (Appendix A). Moreover, CFI-1 samples led to significant TLR4 and TLR2 expression at the concentration of 50 µg/mL. They accounted for approximately 48.0% and 29.1% of control ligand responses, respectively (Appendix A). However, CFI-1 samples did not induce significant human TLR5, TLR7, TLR8, or TLR9 expression (Appendix A). Nonetheless, they had a minor expression effect on TLR3 at the concentration of 50 µg/mL and corresponded to approximately 9.4% of control ligand responses (Appendix A).

### 2.2. Patients’ Baseline Demographics and General Features

In total, 44 patients who met the inclusion criteria were subjected to treatment with OncoTherad^®^ (MRB-CFI-1). Among them, 30 (68.2%) were men and 14 (31.8%) were women, with a mean age of 65 years (Table 1). With respect to bladder cancer risk factors, 31.8% of patients were smokers, 52.3% were former smokers, and 13.6% were non-smokers (Table 1). In addition, one patient reported to have been continuously exposed to aniline in a paint factory (Table 1).

Most patients (84.1%) had previously undergone two courses of intravesical BCG therapy, as shown in Table 1. Nine patients (20.4%) had previously undergone intravesical chemotherapy with gemcitabine either after BCG therapy failure or without it (Table 1). When it comes to BCG responses, 59.1% of patients presented a refractory response type, 31.8% experienced disease relapse, and 9.1% were deemed intolerant to it (Table 1).

With respect to initial disease staging and histological grading after BCG and/or intravesical chemotherapy treatment, tumors’ prevalence was categorized as high-grade pTa (56.8%, Figure 2E,F), which was followed by high-grade pT1 (32.4%, Figure 2G,H) and pTis (10.8%, Figure 2C,D) (Table 1). A multifocal tumor was observed in 63.6% of patients, whereas single lesions were found in 36.4% of cases (Table 1). Furthermore, tumors larger than 3 cm were identified in 81.8% of patients, whereas tumors smaller than 3 cm were detected in 18.2% of cases (Table 1).

The mean number of relapses after prior BCG and/or intravesical chemotherapy with gemcitabine was 2.2 relapses per patient (Table 1).

### 2.3. OncoTherad^®^ (MRB-CFI-1) Nanoimmunotherapy Has Shown High Pathological Complete Response (pCR) Rates, as Well as Increased Relapse-Free Survival (RFS) and Response Duration (RD), within a 24-Month Follow-Up

None of the investigated patients experienced disease relapse during initial follow-up (from 1 to 6 months) after OncoTherad^®^ (MRB-CFI-1) treatment application, as indicated in Table 2. However, a subset of patients (27.3%) experienced relapses during the 24 month follow-up duration: 8.3% of relapse cases were observed at the 7-to-9-month follow-up; 41.7% of cases were observed during the 10-to-12-month follow-up; 33.3% were observed during the 13-to-18-month follow-up; and 16.7% were observed during the 19-to-24-month follow-up (Table 2).

Furthermore, based on the comparison between outcomes observed for OncoTherad^®^ (MRB-CFI-1) nanoimmunotherapy and those observed for prior treatments involving BCG and/or intravesical chemotherapy, it was possible to see statistically significant differences (*p* < 0.0055) * in relapse incidence reductions in patients treated with OncoTherad^®^ (Table 2 and Table 3). Most importantly, no statistically significant disparity in relapse rates was detected during different follow-up periods after OncoTherad^®^ (MRB-CFI-1) treatment application (*p* > 0.05) * (Table 2).

With respect to histological staging and grading after OncoTherad^®^ (MRB-CFI-1) treatment application, relapses were observed across different categories, namely, 66.7% of cases were low-grade pTa, 16.7% were high-grade pT1, 8.3% were high-grade pTa, and 8.3% were pTis tumors (Table 2). Furthermore, OncoTherad^®^ (MRB-CFI-1) nanoimmunotherapy significantly reduced histological tumor grading in comparison to prior treatments involving BCG and/or intravesical chemotherapy (Table 3). More specifically, 100% of patients treated with BCG and/or intravesical chemotherapy presented high-grade tumors, whereas 66.7% and 33.3% of patients subjected to OncoTherad^®^ (MRB-CFI-1) treatment presented low-grade and high-grade tumors, respectively (Table 3).

Likewise, OncoTherad^®^ (MRB-CFI-1) nanoimmunotherapy enabled a substantial reduction in lesion size and focality in comparison to outcomes observed for previous treatments with BCG and/or intravesical chemotherapy (Table 3). In most cases (66.7%), patients experiencing relapse after OncoTherad^®^ (MRB-CFI-1) treatment presented lesions smaller than 3 cm, and these relapses were mostly single lesions.

In total, 32 patients (72.7%) subjected to OncoTherad^®^ (MRB-CFI-1) treatment achieved pCR after the 24-month follow-up (Table 2). The Kaplan–Meier curve plotted for RFS indicated a mean RFS of 21.4 months (equivalent to 642 days) during the 24-month follow-up of patients treated with OncoTherad^®^ (MRB-CFI-1) (Table 2 and Appendix A). Moreover, the mean RD was 14.3 months (Table 2).

### 2.4. OncoTherad^®^ (MRB-CFI-1) Nanoimmunotherapy Promoted Non-Neoplastic Cystoscopic and Tissue Changes in Patients’ Urinary Bladder at the End of the 24-Month Follow-Up

Initial ultrasound and/or cystoscopic examinations evidenced neoplastic lesions >3 cm in the urinary bladder of most patients (81.8%; Table 1 and Table 3) who had undergone previous treatment with BCG and/or intravesical chemotherapy (Figure 2A–H).

At the end of the 24-month follow-up of patients treated with OncoTherad^®^ (MRB-CFI-1), 6.8% (Appendix A) of them showed normal ultrasound (Figure 3A), cystoscopy (Figure 3B), and histology (Figure 3C) results. Non-neoplastic tissue alterations, such as follicular cystitis (Figure 3J–L), discrete chronic cystitis (Figure 3D,E), erosive cystitis with granulation tissue (Figure 3G–I), and flat urothelial hyperplasia with reactive atypia (Figure 3F) were observed in 79.6%, 6.8%, 4.5%, and 2.3% of patients, respectively (Appendix A).

Chronic cystitis (Figure 3D,E) was histologically featured by the presence of lymphoplasmacytic infiltrate and scarce reactive urothelial atypia in response to inflammation. Flat urothelial hyperplasia with reactive atypia (Figure 3F) was histologically featured by thickening and reactive urothelial atypia, focal chronic non-specific inflammatory infiltrate, edema, and mucosal hyperemia.

Erosive cystitis with granulation tissue was featured in cystoscopy by the presence of short and scattered projections (Figure 3G) and histologically featured by the presence of lymphoplasmacytic inflammatory infiltrate with granulation tissue fragments (Figure 3H,I).

Follicular cystitis was identifiable based on the observation of multiple scattered flat lesions on the bladder lining surface during cystoscopy, as shown in Figure 3J. In histological terms, bladder tissues presented several primary and secondary lymphoid follicles. Secondary follicles stood out due to the presence of germinal center-presenting lymphocytes, as well as centrocytes (small and cleaved cells), centroblasts (large cells with prominent nucleoli), and macrophages, as shown in Figure 3K,L.

Therefore, one of the primary outcomes of OncoTherad^®^ (MRB-CFI-1) nanoimmunotherapy relied on stimulating lymphoid follicle formation, a fact that may be linked to its immune-protective effects.

### 2.5. OncoTherad^®^ (MRB-CFI-1) Nanoimmunotherapy Induced Mild-to-Moderate Side Effects during the 24-Month Follow-Up

Thirty-four (*n* = 34; 77.3%) of the herein assessed 44-patient cohort experienced adverse reactions attributable to OncoTherad^®^ treatment, whereas 10 patients (22.7%) remained free from any discernible adverse effects, as shown in Figure 4.

Grade 1–2 adverse reactions were significantly more prevalent than the grade 3–4 ones, as shown in Figure 4 and outlined, in detail, in Table 4. More specifically, 28 patients (63.7%) exclusively presented grade 1–2 adverse reactions, whereas six patients (13.6%) experienced both grade 1–2 and 3–4 adverse reactions (Figure 4). Dysuria, itching, cystitis, arthralgia, fatigue, skin rash, and fever were the grade 1–2 adverse reactions (≥20% incidence) most reported by patients (Table 4). Conversely, the most common grade 3–4 adverse reactions (≥4% incidence) comprised skin rash, diarrhea, cough, and shortness of breath (Table 4).

Furthermore, comprehensive biochemical serological analyses did not show significant differences in parameters such as glucose, hemoglobin, leukocyte count, and platelet count, as well as AST, ALT, GGT, urea, and creatinine levels, before and after OncoTherad^®^ (MRB-CFI-1) treatment application (Appendix A). Consequently, these findings provided evidence that OncoTherad^®^ (MRB-CFI-1) administration at the prescribed therapeutic dosage did not show systemic toxicity indications.

### 2.6. OncoTherad^®^ (MRB-CFI-1) Nanoimmunotherapy Triggered Innate Immune System Activation through TLR4-Mediated Mechanisms and Led to Interferon Signaling Pathway Augmentation, which Was Followed by CD8^+^ T Cell Activation

OncoTherad^®^ (MRB-CFI-1) induced lymphoid follicles’ formation (Figure 3J–L) in most patients (79.6%—Appendix A) after the 24-month follow-up. Therefore, immunoreactions and immunohistochemical analyses conducted after the 24-month follow-up of patients treated with OncoTherad^®^ (MRB-CFI-1) have prioritized regions showing lymphoid follicles (LFs), given the significance of LF formation in bladder tissue.

Immunoreactivity observed for TLR4 (Figure 5A,B) was significantly higher in bladder biopsies conducted after OncoTherad^®^ (MRB-CFI-1) treatment application than in pre-treatment biopsies (Figure 5A,B; Table 5).

Similarly, immunoreactivity levels observed for antigens TRIF (Figure 5C,D), TBK1 (Figure 5E,F), and IRF3 (Figure 5G,H), which are integral components of the TLR4-mediated interferon signaling pathway, recorded statistically significant increases in bladder biopsies conducted after OncoTherad^®^ (MRB-CFI-1) treatment application in comparison to pre-treatment biopsies (Table 5). Moreover, immunoreactivity observed for IFN-γ (Figure 5I,J) recorded significant increases in post-OncoTherad^®^ treatment bladder biopsies in comparison to pre-treatment biopsies (Table 5).

A significantly augmented interferon signaling pathway was observed after OncoTherad^®^ (MRB-CFI-1) treatment application, and it notably increased CX3CR1 (Figure 6A,B) and iNOS (Figure 6I,J) immunoreactivity in opposition to values recorded for pre-treatment biopsies (Table 5). These findings have indicated that this nanoimmunotherapy type induced a cytotoxic immune response.

In order to investigate the potential correlation between CX3CR1 protein levels and individuals’ responses to intramuscular administration of OncoTherad^®^ (MRB-CFI-1) nanoimmunotherapy, these levels were assessed in the peripheral blood of 20 patients before and after treatment application. CX3CR1 protein levels were significantly higher after OncoTherad^®^ (MRB-CFI-1) treatment application (Figure 7 and Appendix A). These findings suggest that this immunotherapy was effective in promoting CD8^+^ T cells’ effector phase.

### 2.7. OncoTherad^®^ (MRB-CFI-1) Nanoimmunotherapy Reduced the Immunosuppression State, as Well as the Immune Checkpoint Immunoreactivity, within the Tumor Microenvironment

Immunoreactivity observed for the immune checkpoint molecules PD-L1 (Figure 6E,F) and CTLA4 (Figure 6G,H) significantly decreased in bladder biopsies conducted after OncoTherad^®^ (MRB-CFI-1) treatment application in comparison to values recorded for pre-treatment biopsies (Table 5).

Similarly, immunoreactivity observed for FOXP3 (Figure 6C,D) significantly decreased in bladder biopsies conducted after OncoTherad^®^ (MRB-CFI-1) treatment application in comparison to values recorded for pre-treatment biopsies (Table 5). Based on this finding, this nanoimmunotherapy type was capable of reversing the immunosuppression state within tumor microenvironment.

## 3. Discussion

NMIBC is a diverse condition featuring varying progression and relapse rates which are influenced by several clinical and pathological factors [10,38,39,40]. Intravesical BCG therapy is widely acknowledged as effective against this condition, and this is the reason why it was the initial treatment adopted to reduce the risk of relapse and progression in high-grade NMIBC cases [10,38,39,40]. Approximately 70% of patients can achieve complete remission depending on their risk profile. However, up to 60% of them may experience relapse within a year [8,40]. Approximately 20% of high-risk cases may progress to muscle-invasive disease within four years despite the BCG treatment application [40,41].

A critical aspect to be taken into consideration in cases of NMIBC relapse, despite adequate BCG therapy application, lies in whether there is a large enough window of time to explore alternative treatment options before contemplating radical cystectomy. Evidence available in the literature has suggested that most patients have approximately one year before experiencing prognosis decline [42]. Moreover, a study did not find significant differences in five-year overall or cancer-specific survival rates between patients participating in clinical trials and those who chose the option for cystectomy right away [43]. However, it is noteworthy that patients presenting any lymphovascular invasion or prostatic urethral involvement degree tend to have poorer prognosis and may not be suitable candidates for second-line therapy.

In cases where intravesical treatment options prove ineffective, the recommended course of action is radical cystectomy with intestinal diversion. While radical cystectomy has the potential to halt local disease progression, it is accompanied by a substantial risk of postoperative complications, a diminished quality of life for patients, and the potential for overtreatment of individuals whose disease may not be significantly advanced. Furthermore, not all patients are suitable candidates for cystectomy due to factors such as advanced age or the presence of severe comorbidities. Considering the life-altering consequences and the adverse impact on quality of life associated with radical cystectomy, a significant number of patients opt to decline this surgical procedure. Consequently, there exists an urgent need for the development of innovative therapies that can offer a non-surgical second-line treatment option.

The BCG non-response point is a crucial time for patients. It is well-established that a third BCG therapy course provides minimal benefit and means the option for undergoing aggressive second-line therapy over cystectomy can potentially lead to tumor progression, among other unfavorable outcomes. On the other hand, radical cystectomy is associated with increased comorbidity rates and may represent over-treatment. Therefore, it is imperative to engage in comprehensive and careful discussions with patients before making these life-changing decisions. Regrettably, guidelines provided by the American Urological Association (AUA) provide limited guidance in this context. Given the scarcity of robust data, many experts in the NMIBC field see cystectomy as the safest option for patients with BCG-non-responsive high-grade NMIBC and for those with high-grade NMIBC who are unable to tolerate full-dose BCG induction and maintenance therapy.

In addition to challenges related to BCG’s toxicity and therapeutic ineffectiveness in high-grade NMIBC patients, the ongoing global shortage in BCG production, as well as its distribution disruptions, contribute to significantly worsening patients’ prognoses. The current BCG shortage has forced clinics to implement rationing and prioritization strategies, and it may have contributed to increase disease relapse and progression rates among NMIBC patients [44,45,46]. Given the aforementioned BCG shortage, patients have been subjected to reduced BCG therapy courses, a fact that may increase the number of patients undergoing cystectomy [44]. Moreover, the BCG shortage has also affected patients’ access and eligibility to participate in clinical trials [44].

Only two NMIBC treatments were approved by the United States Food and Drug Administration (FDA) in the past three decades. Valrubicin, which is classified as anthracycline, was granted FDA approval in 1998. It was followed by pembrolizumab, which is an immune checkpoint inhibitor immunotherapy approved in 2020 [47,48]. NMIBC research progress has been hampered by a lack of consensus concerning clinical trial endpoints and suitable control groups, as well as by challenges associated with enrolling patients in early-stage clinical trials [49]. However, in 2016, the International Bladder Cancer Group (IBCG) introduced guidelines to guide single-arm trials focused on assessing new therapies in cases lacking well-accepted control groups. Subsequently, these guidelines were endorsed by the FDA [10,50]. This governmental agency recommends complete response (CR) as the primary efficacy endpoint for treatment, as well as a follow-up schedule comprising assessments conducted every three months in the first year after treatment, semi-annually in the second year, and on a yearly basis from the third year onwards [51]. On the other hand, the IBCG recommends using CR for carcinoma in situ (CIS) tumors and relapse-free survival (RFS) for high-grade papillary tumors as primary efficacy endpoints [50]. These guidelines have encouraged the development of new studies focused on investigating the use of immunotherapies to treat BCG-unresponsive NMIBC patients. The current study involving OncoTherad^®^ (MRB-CFI-1) immunotherapy adheres to recommendations made by both the FDA and IBCG. It used CR and RFS as primary endpoints, along with a follow-up schedule involving quarterly clinical assessments in the first year and semi-annual assessments in the second year.

Nowadays, second-line treatment options available for BCG-unresponsive NMIBC cases are less than ideal. Although several conventional chemotherapeutic agents, such as gemcitabine, mitomycin, gemcitabine combined with mitomycin, docetaxel, and valrubicin, have been explored, none of them was superior to BCG and remains under investigation. Studies focusing on bladder preservation therapies after BCG relapse reported RFS rates lower than 25% after two years based on single-agent intravesical instillations of chemotherapy and biological preparations. Chemotherapy combinations, most often gemcitabine combination with mitomycin C or docetaxel, recorded RFS rates equal to or lower than 35% after two years [52]. Valrubicin, which is an intravesical analog of doxorubicin and the sole drug approved by FDA for this purpose, recorded a one-year disease-free rate in only 8% of patients [53]. Off-label second-line options comprise interferon-alpha (one-year disease-free rate of 12%) [54,55], gemcitabine (21%) [56], and mitomycin C (23%) [57]. However, if one takes into consideration that several studies reported five-year disease-free rates ranging from 20% to 40% in patients subjected to radical cystectomy, it is quite clear that cystectomy remains the safest option in these cases.

The bladder cancer treatment scenario has significantly evolved over time, since it not only encompasses traditional approaches, such as surgery and chemotherapy, but also immunotherapy integration. BCG immunotherapy has been applied to prevent NMIBC relapse and progression for more than four decades. However, recent improvements in scientists’ understanding about this topic have opened new room to enhance its anti-tumor effectiveness. Alongside BCG, the introduction of immune checkpoint inhibitors has revolutionized the standard care provided to patients with metastatic bladder cancer, as well as expanded their potential use in BCG-unresponsive NMIBC cases. These immune checkpoint inhibitors, such as anti-PD1 or anti-PD-L1 agents, have garnered increasing interest within the bladder cancer-related scientific community. Critical considerations comprise whether using immune checkpoint inhibitors can delay or change patients’ need of undergoing surgery and potentially affect their outcome, as well as whether toxicity is proportional to NMIBC severity.

Pembrolizumab (an anti-PD1 agent) has recently received FDA approval to be applied to high-risk BCG-unresponsive NMIBC patients with and without concomitant CIS who are not suitable candidates for cystectomy. This approval was based on findings resulting from the ongoing phase 2 KEYNOTE-057 study, which achieved a three-month CR rate of 40.6% and a median RD of 16.2 months after more than two years of follow-up with 96 patients. In total, 46.2% of all 39 patients who achieved CR presented RD equal to or longer than 12 months [40,58]. Nine (9) of 57 non-responders experienced disease progression at three months, whereas three out of 36 patients who underwent cystectomy showed disease progression to T2 bladder cancer at least [58].

According to the phase 2 SWOG S1605 monotherapy study conducted with 74 CIS patients treated with atezolizumab (anti-PD-L1 agent), CR rates reached 41.1% at three months and 26.0% at six months [59]. However, this trial was terminated because it did not meet its primary endpoint. Preliminary results of the ongoing phase-2 QUILT-3.032 study enabled the FDA to grant innovative therapy designation to the ALT-803 + BCG IL-15 receptor superagonist in BCG-unresponsive NMIBC patients [60]. In total, 90% of the herein assessed 20 CIS patients achieved CR, whereas 16 high-grade pTa/pT1 patients recorded a 75% survival rate at six months and 54% at nine months [60].

Vicinium (oportuzumab monatox) is a fusion protein composed of an epithelial cell adhesion molecule-specific antibody fragment and Pseudomonas exotoxin. It exerts its antitumor effects by inhibiting protein synthesis [61]. Phase 1 and 2 studies involving intravesical vicinium application reported CR rates ranging from 29% to 40% at three months in patients with CIS [61]. The recently concluded phase 3 VISTA study enabled vicinium fast-track designation by the FDA. Initial phase results have indicated a 40% CR rate at three months, while 52% of patients maintained CR for 12 months or longer [62,63].

Intravesical gene therapy is a promising alternative to treat NMIBC, given the bladder’s suitability for such a treatment. Nadofaragene firadenovec, which is a gene-mediated intravesical therapy based on using human IFN-α2b, recorded a CR rate of 53.4% at three months in a phase 3 study conducted with CIS patients—24.3% of them remained free from high-grade relapse after one year [64]. In total, 72.9% and 43.8% of patients with single high-grade pTa/pT1 tumors maintained relapse-free status at three and twelve months, respectively—patients’ responses to the adopted treatment extended up to one year [64].

OncoTherad^®^ (MRB-CFI-1) nanoimmunotherapy, which provides innovative therapeutic advantages in comparison to conventional treatments, has emerged as promising candidate to help managing high-risk NMIBC, mainly in cases of unsuccessful BCG therapy. Extensive prior research efforts made by our group have evidenced a range of beneficial therapeutic properties and effects associated with OncoTherad^®^ (MRB-CFI-1) [21,22,23,24,25,26,27,28,36]. OncoTherad^®^ (MRB-CFI-1) used in preclinical trials has shown approximately 80% antitumor effectiveness, as well as presented clear superiority over BCG in comparative effectiveness and safety studies [21,22,23,24,25,26,27,28,36].

These promising preclinical effectiveness findings have encouraged our research team to perform a clinical-veterinary trial with dogs diagnosed with naturally occurring bladder cancer. Dogs participating in this trial were treated with intravesical (44 mg/mL) and intramuscular (22 mg/mL) OncoTherad^®^ (MRB-CFI-1) applications on a weekly basis for six consecutive weeks. Subsequently, they underwent intravesical and intramuscular applications of it twice a week for six months, then intravesical and intramuscular applications on a monthly basis for an additional six months (24 applications in total). It is noteworthy that 83.3% of patients presented stabilized disease, whereas 16.7% of them presented partial disease remission after the initial three OncoTherad^®^ (MRB-CFI-1) applications. All (100%) patients achieved partial disease remission after the 24th OncoTherad^®^ (MRB-CFI-1) application. The median relapse-free/progression-free survival time was 402 days. Moreover, OncoTherad^®^ (MRB-CFI-1) administration did not show indications of systemic toxicity at the prescribed therapeutic doses [21,22,29,30,31,32,65].

Another prospective multicenter clinical-veterinary study was conducted with 19 dogs diagnosed with recurrent oral melanoma who had previously undergone at least one course of first-line chemotherapy [33]. OncoTherad^®^ (MRB-CFI-1) treatment based on intramuscular applications (22 mg/mL) administered twice a week for three months was applied to Group G1, which comprised 10 dogs. Subsequently, treatment frequency was reduced to applications every 15 days until the end of the 12-month treatment. Group G2 (nine dogs) was treated with OncoTherad^®^ (MRB-CFI-1) in combination with chemotherapy (Carboplatin: 250–300 mg/m^2^—intravenous) or electrochemotherapy (Bleomycin: 15,000 IU/m^2^—intravenous). Based on treatment outcomes assessed through iRECIST study criteria, complete response occurred in 31.6% of cases in total (Group G1—40.0%; Group G2—22.2%), whereas partial response was observed in 42.1% of cases (Group G1—30.0%; Group G2—55.6%), and stabilized disease was observed in 10.5% of patients (Group G1—20.0%; Group G2—0.0%). Only 15.8% of all patients presented disease progression (Group G1—10.0%; Group G2—22.2%). The median overall progression-free survival time was 640 days (Group G1—676.5 days, Group G2—599.4 days). Therefore, OncoTherad^®^ (MRB-CFI-1) immunotherapy was effective in managing chemotherapy-resistant disease and presented potential advantages in preventing canine oral melanoma progression [33].

The current study recorded a pCR rate of 72.7% (95% CI) and a mean RFS of 21.4 months. The mean response duration among all 32 patients with pCR was 14.3 months. No patient progressed to muscle-invasive or metastatic disease during treatment. Altogether, these findings have emphasized the highly promising antitumor effects of OncoTherad^®^ (MRB-CFI-1) immunotherapy, which surpassed outcomes observed for standard BCG therapy and for other therapeutic regimens used for BCG and/or intravesical chemotherapy non-responsive NMIBC cases. This factor turns OncoTherad^®^ (MRB-CFI-1) immunotherapy into a valuable therapeutic option in NMIBC management.

BCG toxicity can be classified into two categories: local and systemic side effects. Local side effects are more prevalent but, oftentimes, less severe. Sometimes, they can lead to treatment discontinuation [66]. It is noteworthy that most local side effects have a temporary nature [67]. Local side effects’ incidence remains relatively consistent, whether patients are on maintenance therapy or not. These effects often include lower urinary tract symptoms (observed in 57–71% of patients undergoing maintenance therapy and in 38–59% of those who are not undergoing it) and hematuria (observed in 20% of patients undergoing maintenance therapy and 29% of those who are not undergoing it) [68].

Similarly, studies did not identify significant differences in systemic side effects between non-maintenance and maintenance therapy groups [69]. Some of them have even reported reduced toxicity after the initial six-month treatment [66]. Systemic side effects can be categorized into infectious (e.g., bacterial cystitis, epididymitis, prostatitis, and systemic infection) and non-infectious (i.e., arthralgias, skin reactions, and anaphylaxis). Among the most often reported systemic toxicities, one finds fever, chills and flu-like symptoms that affect 22% to 30% of patients undergoing maintenance therapy, as well as 19% to 26% of non-maintenance therapy patients. Epididymitis, prostatitis, and urethral infections were reported in 4% of patients undergoing maintenance therapy and in 4% of non-maintenance therapy patients, whereas systemic infection was observed in 7% of patients undergoing maintenance therapy and in 1% of non-maintenance therapy patients. Less severe systemic toxicity symptoms, such as fever higher than 39.5 °C, rash, arthralgias, and BCGitis (established BCG infection with manifestations in another organ) recorded incidence rates ranging from 2% to 6% [67,68].

Adverse effects observed for pembrolizumab used as a monotherapy were assessed in a cohort of 102 patients. Results have shown that 97.1% of patients experienced at least one adverse event, 65.7% of them had treatment-related adverse events, and 29.4% of the investigated population experienced grade 3–5 adverse events [70].

With respect to OncoTherad^®^ (MRB-CFI-1) nanoimmunotherapy’s safety profile, the current findings have evidenced that treatment-related adverse events (TRAEs) were reported by 77.3% of patients, whereas 22.7% of them did not experience any adverse event. Grade 1–2 TRAEs were the most prevalent types; they were reported by 63.7% of patients in opposition to grade 3–4 events. The most often reported reactions comprised cystitis, dysuria, arthralgia, itching, skin rash, fatigue, and fever. Furthermore, serum levels of several biochemical parameters remained within the normal reference values after OncoTherad^®^ (MRB-CFI-1) treatment application.

Previous studies conducted by our research team have evidenced this nanocompound’s low toxicity in vitro in grade II urinary bladder carcinoma cells (5637 cell line), as well as 75% cell viability, and suggested that neoplastic cell death induced by OncoTherad^®^ (MRB-CFI-1) likely depends on immune system activation [27]. Moreover, the aforementioned studies have shown that OncoTherad^®^ (MRB-CFI-1) immunotherapy did not induce either local or systemic adverse effects on rats, mice, and rabbits at a therapeutic dose of 20 mg/kg [21,23,27,29]. However, OncoTherad^®^ (MRB-CFI-1) doses of 50 mg/kg and 100 mg/kg resulted in moderate and intense inflammatory reactions in the bladder, ureter, and kidneys of the tested animal models. Furthermore, OncoTherad^®^ (MRB-CFI-1) treatment at doses of 20 mg/kg, 50 mg/kg, and 100 mg/kg did not induce hepatotoxicity and nephrotoxicity in rats, mice, or rabbits—the maximum tolerated dose was 100 mg/kg [21,23,27,29].

Briefly, the current study has shown that OncoTherad^®^ (MRB-CFI-1) nanoimmunotherapy induced lower intensity and frequency adverse effects in comparison to those associated with standard BCG therapy and with immune checkpoint inhibitor use. This finding highlights both the safety and the effectiveness of OncoTherad^®^ (MRB-CFI-1) immunotherapy as a viable option to aid in managing patients with carcinoma of the high-risk NMIBC type in situ.

The immune microenvironment of the urinary bladder has unique features. On the one hand, the bladder uses several non-specific defense mechanisms to remain free from infections [71]. These strategies involve the presence of a thick mucin barrier, regular urination, and the secretion of antibacterial agents, such as β-defensins and cathelicidin [71]. Conversely, the bladder maintains an immunosuppressive environment to avoid undesired immune responses [71,72]. Furthermore, bladder tumors’ microenvironments are immunosuppressive, besides leading to tumor-infiltrating lymphocytes’ (TILs) deficiency in effectively ruling out tumor cells [73]. Bladder tumors are often associated with increased regulatory T cell (Tregs) levels, as well as with the expression of inhibitory Th1 response cytokines, such as transforming growth factor-beta (TGF-β) and IL-10. Consequently, understanding the strategies used by bladder tumors to evade the immune system is of paramount importance to assist in developing immunotherapies aimed at fighting bladder cancer [74].

Nowadays, TLR agonists are acknowledged as effective immune stimulants with significant potential to be used in immunotherapy applied to different tumor types, including bladder cancer. One of the immune mechanisms accounting for both detecting and controlling tumor cell proliferation lies in TLR4-mediated production of inflammatory cytokines in macrophages [75,76]. There is downregulation of TLR4 expression in tumor tissues in comparison to surrounding or normal tissues. Most importantly, TLR4 expression is positively associated with overall survival (hazard ratio [HR] = 0.38) and cancer-specific survival (HR = 0.15) rates in bladder cancer patients [77]. In addition, low TLR4 expression is mainly observed in high-grade tumors [77].

The non-canonical TLR4 signaling pathway plays a pivotal role in triggering interferon (IFN) production. IFNs have strong antitumor effects since they enable TNF-related apoptosis-inducing ligand (TRAIL) generation. TRAIL is known for its ability to induce cell death in tumor cells [76]. Tumor-suppressive actions performed by the immune system mostly rely on the activities of IFN-γ, which stimulates several antitumor and antiproliferative pathways in both macrophages and neoplastic cell lines [75]. Moreover, IFNs play a key role in early protection against [75] metastasis, as well as in integral parts of mechanisms involving the innate immune system activation for tumor detection and T cell response modulation purposes [78,79,80,81]. CD8^+^ T lymphocytes’ activation against tumors was compromised in mice lacking STING. Studies conducted in vitro by Woo et al. [81] have evidenced that DNA deriving from neoplastic cells triggered IFN production and activated dendritic cells (DCs) through both the STING pathway and IRF3 and indicated the involvement of a cytosolic DNA detection mechanism. DNA extracted from malignant neoplastic cells from models in vivo was detected within host antigen-presenting cells (APCs), which were correlated to IFN production and STING activation [81].

The current study has shown that OncoTherad^®^ (MRB-CFI-1) immunotherapy was capable of activating the innate immune system mediated by TLR4. This finding was consistent with data available in the literature and with our pre-clinical and veterinary clinical findings on bladder cancer. The aforementioned activation led to a substantial increase in the interferon signaling pathway involving TRIF, TBK1, IRF3, and IFN-γ. This activation plays a critical role in stimulating CD8^+^ CX3CR1^+^ T cells, as well as in reducing the number of immune checkpoint molecules (PD-1/PD-L1 and CTLA4) and in reversing individuals’ immunosuppressive state, with emphasis on decreased FOXP3^+^ Tregs within the urinary bladder microenvironment.

The TLR4-mediated interferon signaling pathway activation induced by OncoTherad^®^ (MRB-CFI-1) also has a significant influence on both the formation and organization of lymphoid follicles (LFs) within the bladder tissue. LFs work as the architectural basis for the development of follicular dendritic cell networks, which play a pivotal role in germinal center reactions. These reactions are essential for affinity maturation processes accounting for optimizing antibodies during adaptive immune responses [82]. TLR expression by myeloid lineage cells establishes a critical link between the innate and adaptive immune responses, mainly in dendritic cells and monocyte–macrophage lineage cells accounting for expressing TLRs. This interaction enhances the function of antigen-presenting cells and enables the release of cytokines that play essential roles in activating CD8^+^ and CD4^+^ T cells [82,83]. These findings emphasize the pivotal role played by TLRs in driving processes that ultimately lead to the development of a fully functional cellular immune response. More specifically, TLR4 is implicated in the maturation of follicular dendritic cells, as well as in the establishment of germinal centers within lymphoid follicles [82].

The role played by TLRs in enhancing humoral immune responses, as well as in affecting B cells both directly and indirectly through their interactions with other immune cells, has been increasingly acknowledged. B cells themselves express TLRs. Moreover, they undergo polyclonal stimulation when they are exposed to TLR ligands such as OncoTherad^®^ (MRB-CFI-1). This stimulation leads to B cells’ differentiation and proliferation into cells accounting for secreting immunoglobulins. TLR signaling also plays a crucial role in enabling B lymphocytes’ migration to and clustering within lymphoid follicles. This process promotes sustained B cell proliferation, besides generating memory B and plasma cells [82]. In addition to its direct effects on B cells, TLR signaling indirectly supports T cell-dependent B cell response induction through its cooperative interactions with CD4^+^ T cells. This synergy between TLRs and CD4^+^ T cells contributes to enhanced B cell responses within the immune system.

Although TLRs’ influence on triggering humoral immune responses is well-documented for helper T cells and for the intrinsic activation of B cells, the general understanding concerning the potential consequences of TLR ligands in LFs and follicular dendritic cells remains limited. Consequently, the current study has emphasized the notable influence of the TLR4 signaling pathway activation induced by (MRB-CFI-1) on LFs’ development and arrangement processes. This OncoTherad^®^ phenomenon may be associated with OncoTherad’s immunoprotective effects on bladder tissue.

Identifying blood-based biomarkers capable of reflecting dynamic changes in the tumor microenvironment and of predicting tumor responses to immunotherapies is a promising path for enhancing current treatment strategies [18]. The current study focused on investigating CX3CR1 as a potential predictive biomarker to measure bladder cancer responses to OncoTherad^®^ (MRB-CFI-1) immunotherapy. According to Yamauchi et al. [18], the administration of immune checkpoint inhibitor immunotherapies to tumor-bearing mice has increased both T-cell receptor frequency and clonality within the peripheral CX3CR1^+^CD8^+^ T-cell subset. Furthermore, their study evidenced that the early surge in CX3CR1^+^ subset frequency among circulating CD8^+^ T cells was correlated to both responses by and prolonged survival of non-small cell lung cancer patients undergoing anti-PD-1 therapy. The current findings have indicated that increased CX3CR1 levels, both in bladder tissue and peripheral blood samples, mirrored dynamic fluctuations observed in CD8^+^ TILs within the tumor microenvironment after OncoTherad^®^ (MRB-CFI-1) treatment application. This increase in CX3CR1 levels is associated with anti-tumor responses triggered by OncoTherad^®^ (MRB-CFI-1) therapy applied to BCG-unresponsive patients with carcinoma in situ. Consequently, the current results support CX3CR1 adoption as a biomarker to help in prognosticating clinical responses to OncoTherad^®^ (MRB-CFI-1) therapy.

Our study has certain limitations that should be acknowledged. Firstly, the relatively small sample size may not provide a precise reflection of the safety and efficacy of OncoTherad^®^ (MRB-CFI-1) immunotherapy. It is important to note that our results are derived from a single institution with expertise in NMIBC treatment protocols. However, it is conceivable that these findings may not be readily applicable to other institutions with lower case volumes of NMIBC.

Moreover, our single-arm study was constrained by the absence of a direct comparator group. Nevertheless, it’s worth emphasizing that there are limited effective treatment alternatives available beyond radical cystectomy for this patient population. Additionally, comparing our findings with off-label treatments is a complex endeavor due to historical inconsistencies in defining adequate BCG therapy, variations in BCG dosing regimens, the inherent heterogeneity of patient populations, and differences in monitoring or surveillance protocols.

In recognition of these limitations, our research group is actively pursuing further investigations to substantiate the safety and efficacy of OncoTherad^®^ (MRB-CFI-1) immunotherapy. Our future research initiatives will encompass expansions to multiple clinical research centers and a substantial augmentation of the sample size.

## 4. Materials and Methods

### 4.1. Assessing Toll-like Receptor (TLR) Ligands in Human Cells: OncoTherad^®^ (MRB-CFI-1) and Its Components (CFI-1 and P14-16 Protein)

A series of experiments involving NF-κB activation in HEK293 cells genetically engineered to express specific TLRs was conducted to assess the impact of both OncoTherad^®^ (MRB-CFI-1) and its constituent components (CFI-1 and P14-16 protein) on toll-like receptors (TLRs). An NF-κB activation-responsive promoter was used to control the expression of a secreted alkaline phosphatase (SEAP) reporter gene, and it enabled monitoring of the TLR signaling through NF-κB activation.

A screening test was applied to assess the effects of both OncoTherad^®^ (MRB-CFI-1) and its constituents (CFI-1 and P14-16 protein) on seven different human TLRs, namely, TLR2, TLR3, TLR4, TLR5, TLR7, TLR8 and TLR9. Ligands acknowledged for these TLRs were used for control purposes, as follows:TLR2: Heat-killed *Listeria monocytogenes* (HKLM). Concentration: 10^8^ cells/mL.TLR3: Poly(I:C). Concentration: 1 µg/mL.TLR4: *Escherichia coli* K12 lipopolysaccharide (*E. coli* K12 LPS). Concentration: 100 ng/mL.TLR5: *Salmonella Typhimurium* flagellin (*S. Typhimurium* flagellin). Concentration: 100 ng/mL.TLR7: CL097. Concentration: 1 µg/mL.TLR8: CL075. Concentration: µg/mL.TLR9: CpG ODN 2006. Concentration: 1 µg/mL.NF-κB control cells: TNF-α. Concentration: 100 ng/mL.

Samples of both OncoTherad^®^ (MRB-CFI-1) and its constituents (CFI-1 and P14-16 protein) were prepared by diluting them in 1 mL of dimethyl sulfoxide (DMSO) to achieve a stock concentration of 5 mg/mL. Subsequently, these samples were tested at final concentrations of 0.5, 5, and 50 µg/mL—results were compared to control ligands. Each step of this process was carried out in duplicate. In total, 20 µL of OncoTherad^®^ (MRB-CFI-1) samples or of its constituents (CFI-1 and P14-16 protein), along with positive control ligands, were introduced in 96-well plates—each well contained the total volume of 200 µL and 50,000 cells. Culture medium was added to the wells used to detect NF-κB-induced SEAP expression. Optical density (OD) was measured in a Beckman Coulter AD 340C Absorbance Detector at 650 nm after 16–20 h of incubation.

### 4.2. Pharmacological Treatment Delivered to NMIBC Patients: OncoTherad^®^ (MRB-CFI-1) Nanoimmunotherapy

In total, 58 patients originating at Paulínia Municipal Hospital (HMP), Paulínia City, São Paulo State, Brazil, who had been diagnosed with NMIBC classified as refractory, recurrent, or intolerant to Bacillus Calmette–Guérin (BCG) and/or to intravesical chemotherapy, were included in the current study. Forty-four (44) of them met the predefined inclusion criteria and started being treated with OncoTherad^®^ (MRB-CFI-1). The recruitment phase spanned from August 2018 to June 2021. It was followed by a subsequent treatment and by a 24-month clinical follow-up.

Eligible participants were patients aged 18 years or older, presenting with histologically confirmed high-risk NMIBC, who were either not suitable candidates for or declined radical cystectomy. Hematological and biochemical laboratory assessments were mandatory 14 days prior to the commencement of the study. To meet eligibility criteria, patients underwent transurethral resection of the bladder tumor (TURBT) four weeks prior to the initiation of the study. This procedure was conducted at HMP and was performed under either general or epidural anesthesia, with the choice being determined through consultation between anesthetists and patients. Confirmation of diagnosis through bladder biopsies was carried out by the Pathology Department of HMP (Figure 8).

BCG-unresponsive disease was defined as: (i) BCG refractory: failure to achieve a disease-free state by six months after initial BCG therapy because of persistent or rapidly recurrent disease. This category also includes patients with stage or grade progression at three months despite BCG therapy; (ii) persistent high-risk NMIBC within six months after adequate BCG therapy; (iii) recurrent high-risk NMIBC within nine months after the last BCG instillation, despite adequate BCG therapy; (iv) BCG intolerant: disease recurrence after the administration of a less than adequate course of BCG therapy because of a serious adverse event or symptomatic intolerance. Adequate BCG therapy was defined as a minimum of five induction instillations (adequate induction) and at least seven instillations of BCG within nine months after the first instillation, consistent with the FDA-recommended definition (≥5 out of six doses of an initial induction course plus ≥2 out of three doses of maintenance therapy or ≥2 out of six doses of a second induction course). Eligible patients had not received intervening intravesical chemotherapy or immunotherapy from the time of the most recent cystoscopy or TURBT until study entry.

Patients with histologically confirmed muscle-invasive carcinoma, locally advanced unresectable or metastatic urothelial carcinoma, or concurrent extravesical urothelial carcinoma (in the prostatic urethra, distal urethra, ureter, or renal pelvis) were excluded (Figure 8).

Upon diagnosis, patients provided informed consent and subsequently started treatment with OncoTherad^®^ (MRB-CFI-1). The OncoTherad^®^ treatment regimen comprised two phases (Figure 8):(a)Induction Phase: Intravesical (at concentration of 120 mg/mL, intravesically retained for 1 h, while patients were asked to rotate positions to maximise bladder surface exposure) and intramuscular (at concentration of 25 mg/mL) OncoTherad^®^ (MRB-CFI-1) administrations on a weekly basis for six consecutive weeks.(b)Maintenance Phase: Biweekly OncoTherad^®^ (MRB-CFI-1) administrations (both intravesical and intramuscular) for three months, followed by monthly administrations (both intravesical and intramuscular) for additional nine months, totaling one year of treatment. Subsequently, quarterly OncoTherad^®^ (MRB-CFI-1) administrations (both intravesical and intramuscular) were maintained for another year.

The current study was approved by the Research Ethics Committee/UNICAMP (CAAE: 93619718.7.0000.5404) and registered in the Brazilian Clinical Trials Registry (RBR-6swqd2).

### 4.3. Assessing the Therapeutic Effectiveness of OncoTherad^®^ (MRB-CFI-1) Nanoimmunotherapy

The primary criteria used to assess treatment effectiveness comprised pathological complete response (pCR), relapse-free survival (RFS), and response duration (RD). pCR was defined as the number of patients who did not show any evidence of disease or disease progression. RFS refers to the time interval from the beginning of OncoTherad^®^ (MRB-CFI-1) treatment to either disease relapse or death as an initial event. Relapse was featured as any histologically confirmed tumor recurrence, regardless of its grade. Response duration was defined as the time interval between achieving pCR and experiencing therapeutic failure.

All patients were subjected to regular follow-up consultations at the outpatient clinic of HMP to assess therapeutic effectiveness. This follow-up process encompassed histopathological assessments, urinary tract ultrasounds, cystoscopies, and laboratory examinations every three months during the first follow-up year and, subsequently, every six months during the second year (Figure 8).

### 4.4. Inclusion and Exclusion Criteria

#### 4.4.1. Inclusion Criteria

Patients meeting the following criteria were eligible to be included in the current study:Aged ≥18 yearsHistologically confirmed diagnosis of NMIBC (Ta, T1, and/or carcinoma in situ—CIS) with urothelial carcinoma—predominant histology.Presence of multiple tumors (>2 lesions) and/or recurrent and/or large tumors (>3 cm) classified as TaG1-2.Diagnosis of grade 3 urothelial bladder carcinoma.Completed ≥1 adequate course of BCG induction therapy for the treatment of NMIBC.Persistent or recurrent high-risk NMIBC after adequate induction therapy.Patients who were refractory or intolerant to BCG treatment.Patients who were recurrent or refractory to first and second-line chemotherapy since they are at higher risk of disease progression to invasive and/or metastatic cancer.Willingness to provide informed consent.Clinical ineligibility for radical cystectomy.Patients who declined radical cystectomy.Adequate organ function.

#### 4.4.2. Exclusion Criteria

Patients meeting any of the following criteria were excluded from the current study:Diagnosis of muscle-invasive bladder tumors.Metastatic bladder tumors.Histological types other than urothelial carcinoma, including adenocarcinoma and squamous cell carcinoma.Patients with cardiac function classified as higher than class III since it indicates the need of cardiac procedures or treatments.Presence of comorbidities, such as demyelinating diseases of the central nervous system (e.g., multiple sclerosis, optic neuritis), congestive heart failure class II or higher, chronic obstructive pulmonary disease, interstitial lung diseases (including pulmonary fibrosis), tuberculosis (any form), chronic kidney failure stage II or higher, liver cirrhosis of any cause, chronic viral hepatitis, HIV infection (with or without AIDS), leprosy, systemic lupus erythematosus, rheumatoid arthritis, scleroderma, any glomerular kidney disease, bullous skin diseases, chronic fungal infections, chronic parasitic diseases, or being a transplant recipient on immunosuppressants.Current or prior diagnosis of any other malignant neoplasm (except for non-melanoma skin cancer) unrelated to the inclusion tumor.Having access to any antineoplastic immunotherapy approved to treat their respective tumor.

### 4.5. Histopathological Analysis

Bladder samples collected from all patients through cystoscopy or transurethral resections of bladder tumors (TURBT) were fixed in 10% formaldehyde for 24 h and then embedded in paraffin. Subsequently, they were sliced into 5 μm-thick sections with a semi-automatic microtome (CUT5062; Slee Mainz, Munich, Germany). These sections were stained with hematoxylin–eosin and examined under a Leica DM2500 microscope equipped with a DFC295 camera (Leica, Munich, Germany) in order to capture images.

Diagnosis based on bladder biopsies was initially performed by the Pathology Department of HMP and further reviewed by a senior pathologist from the Medical School at the University of Campinas (UNICAMP). Urothelial lesions were categorized based on the staging criteria proposed by the World Health Organization/International Society of Urological Pathology consensus [84].

### 4.6. Toxicological Analyses of OncoTherad^®^ (MRB-CFI-1) Nanoimmunotherapy

All patients underwent clinical assessments using the Toxicity Scale, as outlined in the 4th Common Terminology Criteria for Adverse Events [85], to investigate potential local and systemic toxicities associated with OncoTherad^®^ (MRB-CFI-1) nanoimmunotherapy. Blood and urine samples were collected every three months during the first follow-up year, as well as every six months during the second year, to perform the following assessments:Complete blood count, including hemoglobin and white blood cell counts.Thrombogram.Serum glucose, aspartate transaminase (AST), alanine transaminase (ALT), gamma-glutamyl transferase (GGT), urea, and creatinine levels.

These assessments were conducted at the Clinical Analysis Laboratory of HMP. Local toxic effects were investigated based on cystoscopy findings, on the incidence of hematuria, as well as on the assessment of lower urinary tract symptoms.

### 4.7. Immunohistochemical Analyses Applied to TLR4, TRIF, TBK1, IRF3, IFN-γ, CX3CR1, FOXP3, PD-L1, CTLA4, and iNOS in Bladder Biopsies, Both before and after OncoTherad^®^ (MRB-CFI-1) Treatment Application

Both pre- and post-treatment bladder biopsy samples (*n* = 44) collected from patients treated with OncoTherad^®^ (MRB-CFI-1) were subjected to immunohistochemical analyses after histopathological diagnosis. In order to do so, bladder tissue samples were sliced into 5-μm thick sections with a semi-automatic microtome, Slee CUT5062 RM 2165 (Slee Mainz, Mainz, Germany), and placed on silanized slides. Then, these sections were subjected to deparaffinization with xylene, rehydrated with graded alcohols, and rinsed with deionized water. Antigen retrieval was carried out using citrate buffer (pH 6.0) in a microwave at 100 °C and at 800 W power for 20 min. This procedure was followed by blocking endogenous peroxidase activity using 0.3% H_2_O_2_ for 15 min. Non-specific protein binding was blocked based on using a 5% goat serum solution in TBS-T buffer at room temperature for 30 min.

Subsequently, the immunolocalization of antigens such as TLR4, TRIF, TBK1, IRF3, IFN-γ, CX3CR1, FOXP3, PD-L1, CTLA4 and iNOS was performed using specific primary antibodies (Appendix A), which were diluted in 1% goat serum and incubated at 4 °C overnight. Antigen detection was achieved using the EasyLink One Polymer HRP IHC kit (EP-12-20504, EasyPath) according to the manufacturer’s guidelines. Bladder tissue sections were exposed to horseradish peroxidase (HRP)-conjugated secondary antibody from the EasyLink One Polymer HRP IHC kit for 40 min, after which they were rinsed with TBS-T buffer. Subsequently, they were visualized using chromogen 3-3′-diaminobenzidine tetrahydrochloride (DAB), counterstained with Harris hematoxylin, and assessed under a Leica DM2500 microscope equipped with a *DFC295* camera (Leica, Munich, Germany).

Ten (10) fields (at 400× magnification) were selected for each patient and antibody to assess antigen immunoreactivity intensity in bladder tissue. Immunostaining results were analyzed in Image J software version 1.54f. Macro profile analysis was used to quantify positive cells [28,36]. Total immunoreactivity was calculated as the rate of negative cells for a specific antibody subtracted from 100%. It represented the total number of cells in the field that were immunoreactive to the assessed antibody [28,36].

### 4.8. Western Blotting Analysis Applied to CX3CR1 in Patients’ Peripheral Blood, Both before and after OncoTherad^®^ (MRB-CFI-1) Treatment Application

Peripheral blood samples were collected from 20 patients, both before and after OncoTherad^®^ (MRB-CFI-1) immunotherapy application. Patients refrained from taking medication 72 h before sample collection in order to minimize potential drug effects on their immune system. Blood drawn from each patient was placed in 3.5 mL tubes (Vacuette—REF 454327, Greiner Bio-One GmbH, Kremsmünster, Austria) using 0.5 mL of 3.2% (*w*/*v*) sodium citrate as an anticoagulant; the blood/sodium citrate ratio was kept at 9:1. Blood samples were centrifuged in a Rotina 380R centrifuge (Hettich Zentrifugen, Munich, Germany) at 3500 rpm at 25 °C for 10 min right after collection. After the centrifugation process was over, plasma was separated from each patient’s sample with a 200 μL micropipette and transferred to 2.0 mL microtubes. Then, 10 μL/mL of protease inhibitor cocktail (MilliporeSigma, Burlington, MA, USA) was added to each microtube filled with plasma. A portion of each sample was used to determine the protein concentration based on the Bradford microplate assay method. Subsequently, 50 micrograms of protein were loaded onto an SDS-polyacrylamide gel for electrophoresis purposes, then they were electrically transferred to nitrocellulose membranes [27].

Nitrocellulose membranes were blocked with 3% bovine serum albumin (BSA) solution diluted in TBS-T buffer to avoid non-specific protein binding. These membranes were incubated with primary antibodies, such as anti-CX3CR1 (dilution: 1:1500 in 1% BSA) and anti-β-actin (dilution: 1:1500 in 1% BSA), at 4 °C overnight (Appendix A). Subsequently, they were incubated with HRP-conjugated secondary antibodies (diluted 1:3000 in 1% BSA; MilliporeSigma, Burlington, MA, USA) for 2 h.

Bands’ immunoreactivity was visualized by incubating them with DAB (diaminobenzidine) chromogen (Sigma Chemical Co., St. Louis, MO, USA). Samples collected from all 20 patients, both before and after OncoTherad^®^ (MRB-CFI-1) treatment application, were grouped into four different pools (each pool comprised *n* = 5 samples). Immunoblots were conducted in duplicate, and semiquantitative densitometric analysis (IOD—Integrated Optical Density) applied to the bands was performed in Image J software version 1.54f. This analysis was followed by statistical assessment, whose results were expressed as the mean ± standard deviation in comparison to β-actin labeling intensity, which was used as an endogenous positive control [27,28].

### 4.9. Statistical Analyses

Categorical variables were summarized in frequency tables, which comprised absolute frequencies (n) and percentages (%), whereas numerical variables were summarized through descriptive statistics (mean, standard deviation, minimum, median, and maximum) to provide an overview of sample features in comparison to the investigated variables.

Tumor size, focus, and frequencies (both absolute and percentage) were tabulated to assess tumor response to OncoTherad^®^ (MRB-CFI-1) treatment with respect to histological grade. Fisher’s exact test was used for statistical analysis purposes.

Quantitative variables were introduced based on central tendency and dispersion measures for the analysis applied to relapse and biochemical serological parameters, both before and after OncoTherad^®^ (MRB-CFI-1) treatment application. Data distribution was assessed based on parameters such as homogeneity of variance and adherence to the normal curve through the Kolmogorov–Smirnov test. Non-parametric tests (specifically the Kruskal–Wallis test) were applied to variables that did not conform to normal distribution.

Descriptive statistics were computed (mean, standard deviation, minimum, median, and maximum) based on the time taken to achieve relapse-free status, more specifically, pathological complete response. This was done in order to assess relapse-free survival rates over time. A relapse-free survival curve was plotted based on the Kaplan–Meier method.

A McNemar test was used to assess the incidence of adverse reactions (categorized as No or Yes) based on their intensity grade (1–2 or 3–4). Descriptive statistics were used to compare the median values recorded for the total number of grade 1–2 reactions to those of grade 3–4 reactions based on the Wilcoxon test. Moreover, each reaction type was assessed separately, while reaction frequencies (both absolute and percentage) were recorded.

Quantitative data resulting from immunohistochemistry and Western blotting tests were expressed as mean ± standard deviation. Parametric analysis of variance (ANOVA) was carried out. It was followed by Tukey testing whenever normality and homoscedasticity assumptions were met. Non-parametric analysis of variance (Kruskal–Wallis test) was used in cases whose data did not follow normal distribution. Student–Newman–Keuls testing was used for post-hoc comparisons.

All statistical analyses were conducted by the Biostatistics Service of the Medical School at UNICAMP in the SAS System for Windows (Statistical Analysis System) version 9.4, as well as in R software version 3.4.2, which was developed by The R Foundation for Statistical Computing.

## 5. Conclusions

OncoTherad^®^ (MRB-CFI-1) treatment effectively activated patients’ innate immune system through TLR4, and it significantly upregulated the interferon signaling pathway. This activation played a crucial role in stimulating CD8^+^ CX3CR1^+^ T cells, while reducing the number of immune checkpoint markers and reversing the immunosuppressive conditions within the urinary bladder microenvironment. Moreover, the TLR4-mediated interferon signaling pathway induced by OncoTherad^®^ (MRB-CFI-1) contributed to both the formation and organization of lymphoid follicles in the bladder tissue. This process has potentially contributed to its antitumor and immune-protective effects.

Therefore, OncoTherad^®^ (MRB-CFI-1) nanoimmunotherapy emerges as a safe and effective treatment option for patients with BCG-unresponsive NMIBC and for those who have undergone intravesical chemotherapy. This therapy has the potential to help reduce tumor relapse rates and potentially delay or rule out the need for conducting radical surgical interventions in these patients.

## 6. Patents

OncoTherad^®^ (MRB-CFI-1) nanoimmunotherapy holds patents granted in Brazil (INPI: BR102017012768B1), in the United States (USPTO: US-11623869-B2; US-11572284-B2; US-11136242-B2; US-11639294-B2), and in Europe (EPO: EP3626746B1; EP3998075B1).

## Figures and Tables

**Figure 1 ijms-24-17535-f001:**
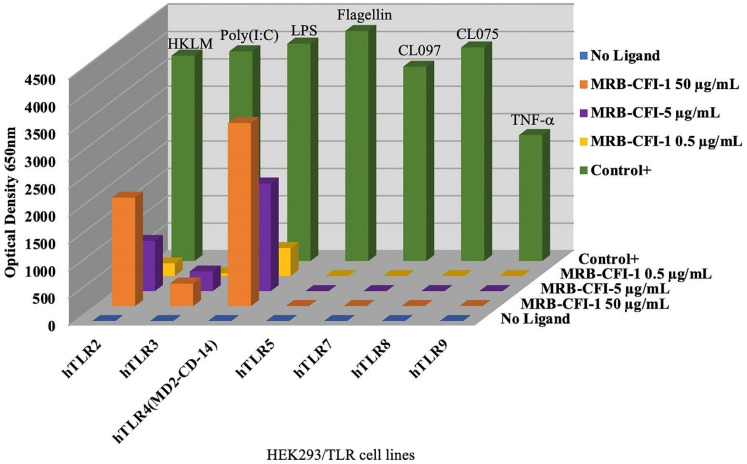
Human TLR ligand screening—OncoTherad^®^ (MRB-CFI-1).

**Figure 2 ijms-24-17535-f002:**
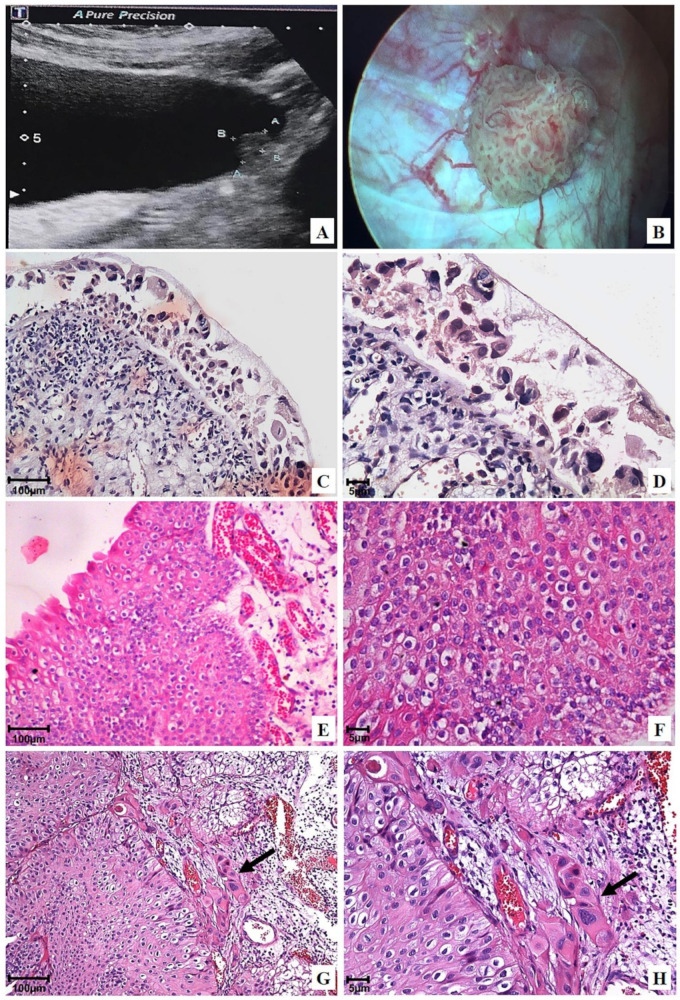
Representative ultrasonography, cystoscopy, and photomicrographs of the urinary bladder before OncoTherad^®^ (MRB-CFI-1) treatment: (**A**) Ultrasonography image revealing an oval-shaped hyperechoic lesion, measuring 3.0 × 1.0 × 0.9 cm in the left lateral wall; (**B**) Cystoscopy image confirming a vegetative lesion >3 cm in the left lateral wall; (**C**,**D**) Flat carcinoma in situ (pTis); (**E**,**F**) Non-invasive papillary urothelial carcinoma (pTa) 2 + 2 (Unicamp) or high-grade (WHO/ISUP); (**G**,**H**) Urothelial papillary carcinoma 2 + 2 (Unicamp) or high-grade (WHO/ISUP) with focal invasion (arrows) of the lamina propria (pT1).

**Figure 3 ijms-24-17535-f003:**
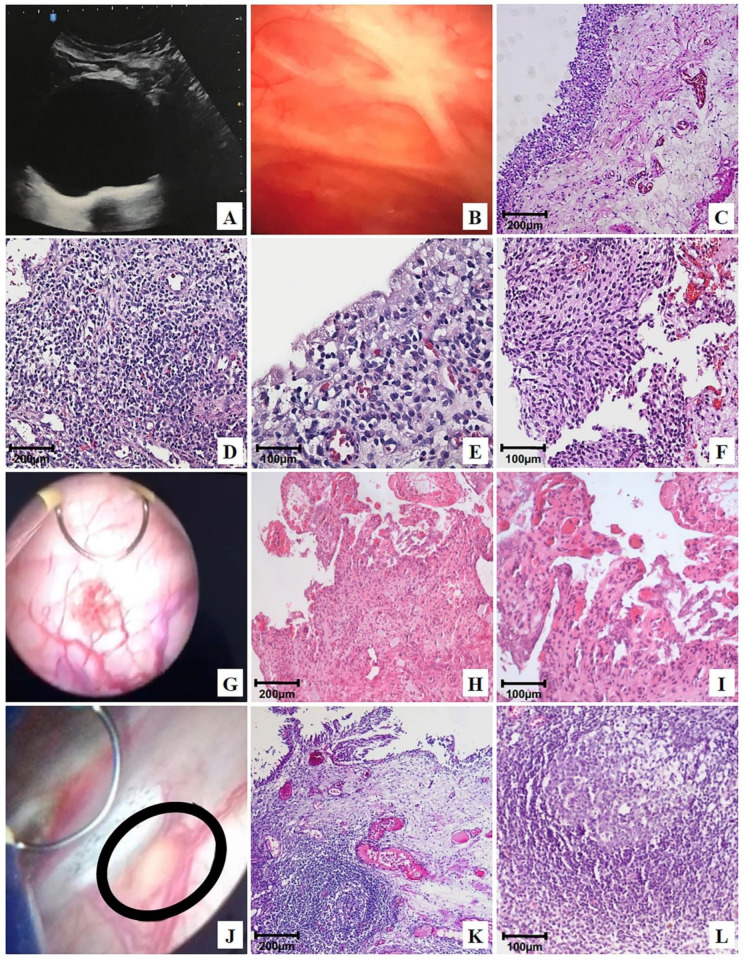
Representative ultrasonography, cystoscopy, and photomicrographs of the urinary bladder after OncoTherad^®^ (MRB-CFI-1) treatment: (**A**) bladder with regular walls and mucosa free of vegetative lesions, showing only the prior (**B**) transurethral resection (TUR); (**C**) normal urothelium and normal lamina propria; (**D**,**E**) mild chronic cystitis; (**F**) flat urothelial hyperplasia with reactive atypia; (**G**) bladder mucosa with lesions exhibiting short, scattered projections, histologically characterized in (**H**,**I**) as granulation tissue (erosive cystitis + granulation tissue); (**J**) cystoscopic image revealing a flat lesion (circle) <3 cm on the bladder floor, characterized as follicular cystitis; (**K**,**L**) photomicrographs of the same flat lesion found during cystoscopy: secondary lymphoid follicle.

**Figure 4 ijms-24-17535-f004:**
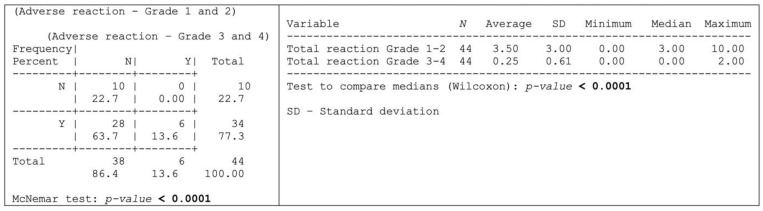
Occurrence of some type of adverse reaction according to the degree of intensity.

**Figure 5 ijms-24-17535-f005:**
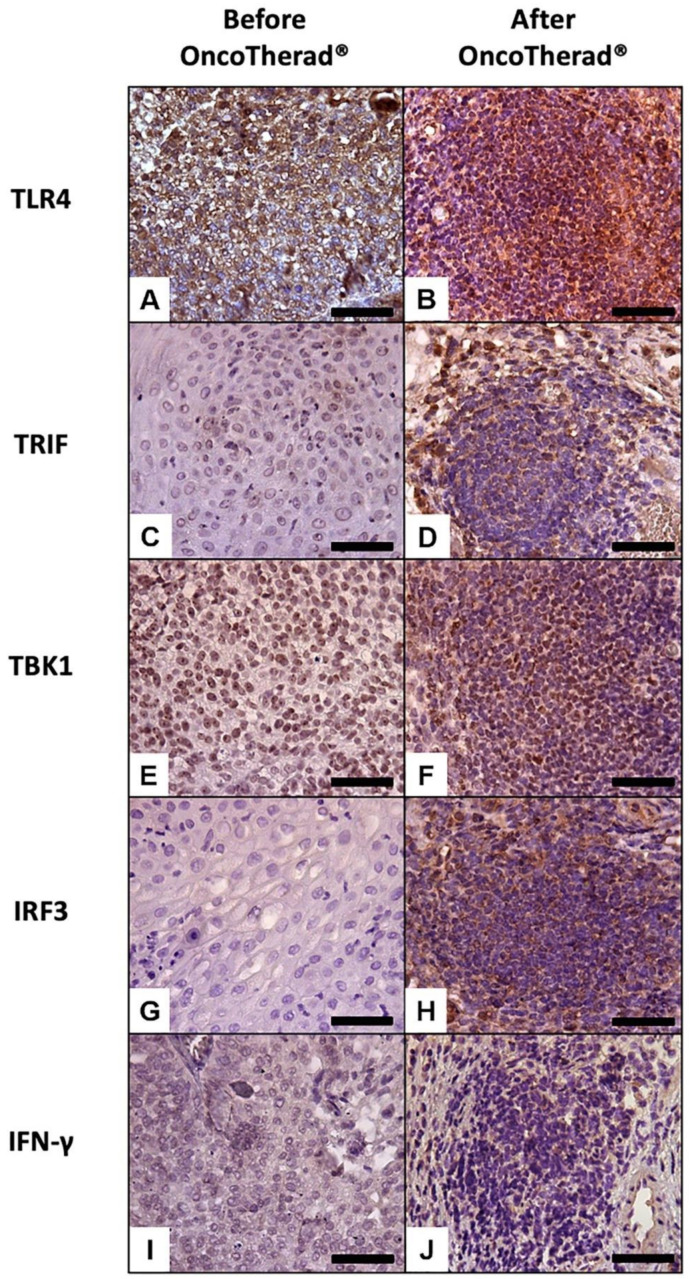
Representative immunolabeling of antigens in the urinary bladder before (**A**,**C**,**E**,**G**,**I**) and after (**B**,**D**,**F**,**H**,**J**) OncoTherad^®^ (MRB-CFI-1) treatment. TLR4 (**A**,**B**); TRIF (**C**,**D**); TBK1 (**E**,**F**); IRF3 (**G**,**H**); IFN-γ (**I**,**J**). Scale bars = 50 μm.

**Figure 6 ijms-24-17535-f006:**
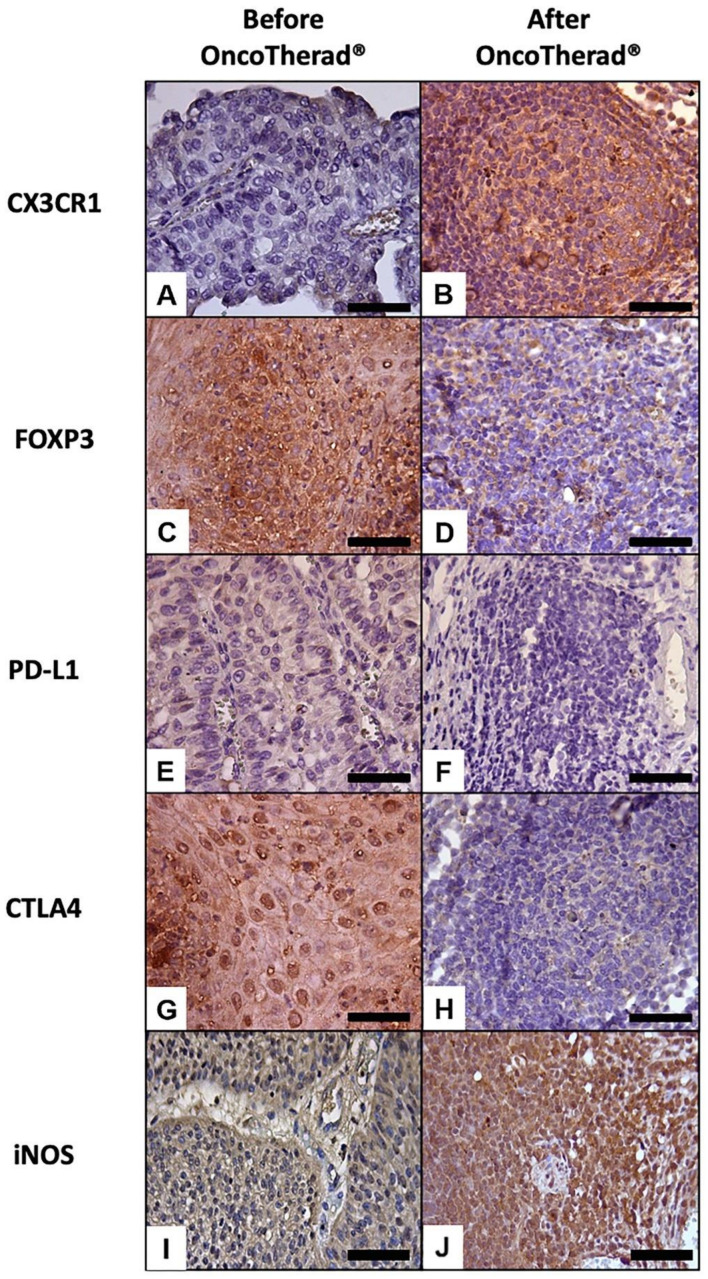
Representative immunolabeling of antigens in the urinary bladder before (**A**,**C**,**E**,**G**,**I**) and after (**B**,**D**,**F**,**H**,**J**) OncoTherad^®^ (MRB-CFI-1) treatment. CX3CR1 (**A**,**B**); FOXP3 (**C**,**D**); PD-L1 (**E**,**F**); CTLA4 (**G**,**H**); iNOS (**I**,**J**). Scale bars = 50 μm.

**Figure 7 ijms-24-17535-f007:**
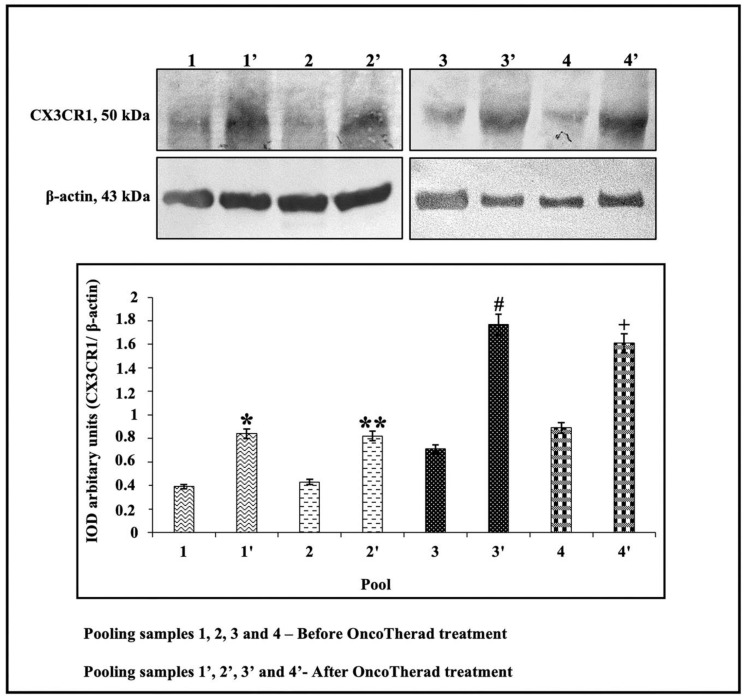
Representative immunoblots and semiquantitative determination of CX3CR1 protein levels. Representative protein profiles grouped from five patients per sample pool: before (pools 1, 2, 3, 4) and after (pools 1′, 2′, 3′, 4′) OncoTherad^®^ (MRB-CFI-1) treatment. Immunoblots were performed in duplicate. The graphs represent the relative integrated optical density (IOD) for CX3CR1 protein, normalized to β-actin and expressed as mean ± standard deviation. *, **, #, + Statistical significance (*p* < 0.01): * Pool 1′ vs. Pool 1; ** Pool 2′ vs. Pool 2; # Pool 3′ vs. Pool 3; + Pool 4′ vs. Pool 4.

**Figure 8 ijms-24-17535-f008:**
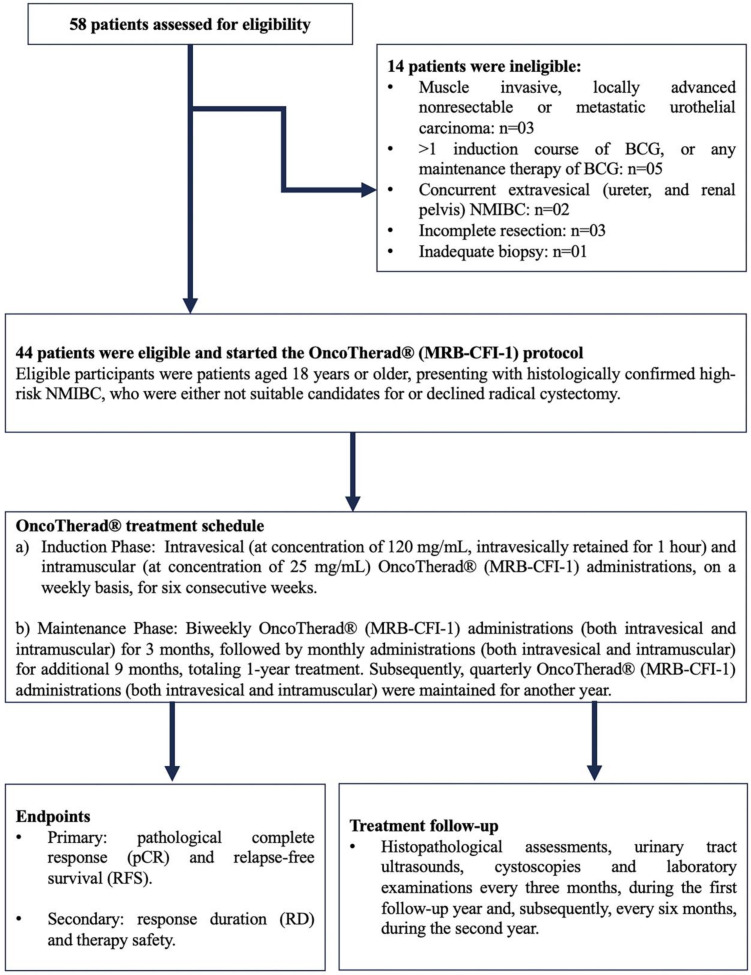
OncoTherad^®^ (MRB-CFI-1) study design. BCG: Bacillus Calmette–Guérin; NMIBC: Non-muscle-invasive bladder cancer.

**Table 1 ijms-24-17535-t001:** Patients’ baseline demographics and general features (*n* = 44).

Characteristics	No. (%)
Age, median (years) (Minimum–Maximum)	65 (34–96)
Sex	
Male	30 (68.2%)
Female	14 (31.8%)
Race	
White	41 (93.2%)
Black or African-American	01 (2.3%)
Asian	02 (4.5%)
Smoking	
Never	06 (13.6%)
Former	23 (52.3%)
Actual	14 (31.8%)
Chronic exposure to aniline	01 (2.3%)
Prior BCG cycles	
1	07 (15.9%)
2	37 (84.1%)
Intravesical chemotherapy	09 (20.4%)
BCG failure	
BCG refractory	26 (59.1%)
BCG relapsing	14 (31.8%)
BCG intolerant	04 (9.1%)
Histological grade after BCG and/or intravesical chemotherapy	
High-grade pT1	14 (32.4%)
High-grade pTa	25 (56.8%)
pTis	05 (10.8%)
Tumor Focality after BCG and/or intravesical chemotherapy	
Multifocal	28 (63.6%)
Single	16 (36.4%)
Tumor size after BCG and/or intravesical chemotherapy	
<3 cm	08 (18.2%)
>3 cm	36 (81.8%)
Number of Relapses after BCG and/or intravesical chemotherapy	
1	04 (9.1%)
2	29 (65.9%)
3	09 (20.5%)
4	02 (4.5%)
Mean (Standard deviation)	2.2 (0.67)

**Table 2 ijms-24-17535-t002:** Descriptive measures of quantitative variables (number of relapses, relapse-free survival, and response duration) and categorical variables (histological grade and pathological complete response) within the 24-month follow-up of patients treated with OncoTherad^®^ (MRB-CFI-1).

Parameters	N	%	Mean	Standard Deviation
Number of Relapses				
1-to-6-month follow-up	0	0	0	0
7-to-9-month follow-up	01	8.3	0.02	0.15
10-to-12-month follow-up	05	41.7	0.11	0.32
13-to-18-month follow-up	04	33.3	0.09	0.29
19-to-24-month follow-up	02	16.7	0.04	0.21
	Absence Frequency = 32			
Total	12	27.3	0.27	0.46
Relapse-Free Survival (RFS)	44	100	21.4	4.8
Response Duration (RD)	12	27.3	14.3	4.1
Histological grade				
pTis	02	16.7		
High-grade pT1	02	16.7		
High-grade pTa	01	8.3		
Low-grade pTa	07	58.3		
	Absence Frequency = 32			
Pathological Complete Response (pCR)	32	72.7		

**Table 3 ijms-24-17535-t003:** Histological grade, tumor size, and tumor focality assessment before (BCG and/or intravesical chemotherapy) and after OncoTherad^®^ (MRB-CFI-1) treatment application.

	BCG and/or Intravesical Chemotherapy		OncoTherad^®^ (MRB-CFI-1)		*p*-Value
Parameters	N	%	N	%	
Histological Grade					
High-grade	44	100	04	33.3 *	0.0055 (Fisher)
Low-grade	0	0	08	66.7 *	
			Absence Frequency = 32		
Tumor Size					0.0055 (Fisher)
<3 cm	08	18.2	08	66.7 *	
>3 cm	36	81.8	04	33.3 *	
			Absence Frequency = 32		
Tumor Focality					0.0055 (Fisher)
Multifocal	28	63.6	04	33.3 *	
Single	16	36.4	08	66.7 *	
			Absence Frequency = 32		
Mean Number of Relapses	2.2	100	0.27	27.3 *	0.0055 (Fisher)
			Absence Frequency = 32		

* Statistical significance.

**Table 4 ijms-24-17535-t004:** Adverse reactions in patients with NMIBC who were subjected to OncoTherad^®^ (MRB-CFI-1) intravesical and intramuscular treatment.

Adverse Reactions	OncoTherad^®^ (MRB-CFI-1) *n* = 44	
	Grade 1–2 N (%)	Grade 3–4 N (%)
General Reactions		
Fatigue	14 (31.8%)	0
Peripheral edema	05 (11.4%)	0
Pyrexia	09 (20.4%)	0
Gastrointestinal Reactions		
Diarrhea	03 (6.8%)	02 (4.5%)
Nausea	06 (13.6%)	0
Vomiting	03 (6.8%)	01 (2.3%)
Constipation	04 (9.1%)	0
Abdominal pain	07 (15.9%)	01 (2.3%)
Urinary Tract Reactions		
Dysuria	23 (52.3%)	0
Cystitis	19 (43.2%)	0
Musculoskeletal and Connective Tissue Reactions		
Arthralgia	15 (34.1%)	0
Skin and Subcutaneous Tissue Reactions		
Pruritus	22 (50.0%)	0
Rash	12 (27.3%)	03 (6.8%)
Respiratory, Thoracic, and Mediastinal Reactions		
Cough	07 (15.9%)	02 (4.5%)
Dyspnea	03 (6.8%)	02 (4.5%)

**Table 5 ijms-24-17535-t005:** Mean ± standard deviation of total immunoreactivity (%) observed for different antigens, both before and after OncoTherad^®^ (MRB-CFI-1) treatment application.

	Groups (*n* = 10 Fields/Group)	
Antigens	Before OncoTherad^®^ Treatment	After OncoTherad^®^ Treatment
TLR4	50.33 ± 5.0 a	95.83 ± 6.6 b
TRIF	47.96 ± 6.9 a	91.24 ± 4.0 b
TBK1	53.92 ± 5.8 a	91.01 ± 3.4 b
IRF3	42.36 ± 6.2 a	90.32 ± 4.5 b
IFN-γ	33.32 ± 3.2 a	82.99 ± 3.7 b
CX3CR1	20.00 ± 5.7 a	93.20 ± 3.3 b
iNOS	31.49 ± 3.7 a	94.94 ± 4.3 b
FOXP3	91.28 ± 6.1 a	34.86 ± 9.8 b
PD-L1	56.78 ± 7.3 a	56.43 ± 4.7 a
CTLA4	92.10 ± 5.0 a	31.98 ± 2.8 b

Values followed by different letters on the same line point out significant differences between groups (*p* < 0.05).

## Data Availability

The data presented in this study are available on request from the corresponding author. The data are not publicly available due to patent issues.

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
