# Peer review of "OncoTherad® (MRB-CFI-1) Nanoimmunotherapy: A Promising Strategy to Treat Bacillus Calmette–Guérin-Unresponsive Non-Muscle-Invasive Bladder Cancer: Crosstalk among T-Cell CX3CR1, Immune Checkpoints, and the Toll-Like Receptor 4 Signaling Pathway"

_ijms, 2023, doi:10.3390/ijms242417535_

Round 1

Reviewer 1 Report

Comments and Suggestions for Authors

The manuscript reports the outcomes of a single-arm phase I/II study employing OncoTherad nanoparticles, developed by the authors, for the treatment of 44 BCG-unresponsive (intolerant, relapsed, and refractory) non-muscle invasive bladder cancer (NMIBC) patients. The nanoparticles induced the expression of TLR4, triggering activation of the innate immune system through TLR-mediated mechanisms, leading to the activation of interferon signaling pathways and subsequent activation of CD8+ T cells. The pathological complete response rate was 72.7% after 24 months, and the mean recurrence-free survival was 21.4 months. No progression to muscle-invasive or metastatic disease was observed during the treatment. Adverse events occurred in 77.3% of the patients, predominantly of grade 1-2 severity.

Hence, this treatment modality appears to be both safe and effective for BCG-unresponsive NMIBC patients. The authors also exposed 293 cells to OncoTherad nanoparticles to assess the expression levels of TLRs.

The findings are effectively communicated through 7 figures and 5 tables, supplemented by 3 additional tables and 4 figures. With 85 references, appropriately incorporated, the paper holds potential interest for a broad audience, including urologists, oncologists, and basic researchers.

To enhance the manuscript further, the authors may consider addressing the following issues:

Tables 1 and 2 are identical. Table 1 should be replaced with one containing the patients' demographic and clinical parameters.

In Table 2, columns for mean, SD, minimum, median, and maximum can be omitted as they lack data. Similarly, in Table 3, the columns for minimum, median, and maximum can be excluded.

Author Response

Ms. Milica Stankovic

Assistant Editor

Manuscript ID: ijms-2736106

Title: " OncoTherad® (MRB-CFI-1) Nanoimmunotherapy: a Promising Strategy to Treat Bacillus Calmette-Guérin-Unresponsive Non-Muscle-Invasive Bladder Cancer: Crosstalk Among T-cell CX3CR1, Immune Checkpoints and Toll-like Receptor 4 Signaling Pathway"

Dear Ms. Milica Stankovic:

Thank you for the e-mail dated December 06, 2023. The reviewers’ comments were of fundamental importance for the data interpretation.

In attention to the recommendations, enclosed can be found a revised manuscript, which has all the corrections indicated by the reviewer. The corrections in the manuscript are highlighted in red. A numbered list indicating how I have dealt with each of the points raised by the reviewers is also provided.

Thank you very much in advance.

Looking forward to reading from you.

Yours sincerely

Prof. Dr. Wagner José Fávaro

Laboratory of Urogenital Carcinogenesis and Immunotherapy (LCURGIM), Department of Structural and Functional Biology, Institute of Biology, University of Campinas (UNICAMP), CP-6109, 13083-865, Campinas, DP, Brazil. Telephone: + 55 (19) 3521-6104. E-mail: [email protected]

In attention to the Reviewer #1

Comments 1: To enhance the manuscript further, the authors may consider addressing the following issues:

Tables 1 and 2 are identical. Table 1 should be replaced with one containing the patients' demographic and clinical parameters.

In Table 2, columns for mean, SD, minimum, median, and maximum can be omitted as they lack data. Similarly, in Table 3, the columns for minimum, median, and maximum can be excluded.

Response 1: We appreciate the reviewer's considerations, which were fundamental in enhancing the interpretation of the data. Table 1 has been extensively revised, and demographic data have been incorporated. Table 2 has been removed.

The Table 3 has been restructured and is now referred to as Table 2.

All manuscript corrections are highlighted in red.

I sincerely hope that all alterations have been made for consider the reviewer’s corrections, and if the paper needs other alterations, please, let me know.

Reviewer 2 Report

Comments and Suggestions for Authors
  • Introduction: Authors should consider elaborating on specific limitations of current BCG treatments to further underscore the need for new therapies.

  • Methods: Authors should consider moving Methods section to before Results section for readability and flow for the reader within the scientific framework.

  • Methods: Provide more details on patient selection criteria to enhance clarity of study population further. Additionally, the authors should consider a more detailed demographics table and expand on the demographic and clinical variable makeup of the patient population beyond the current parameters (age, sex, race, among others)

  • Methods: Authors should consider including a flowchart of inclusion/exclusion criteria and patient selection.

  • Discussion: There should be a section within the discussion that provides an overview of any potential limitations of the study (sample size, methodology, limitations of the therapy itself, etc.) as well as clarified future directions for research and clinical applications.

  • Intro and Discussion should include history of BCG and an overview of prior therapies to set the background for this novel therapy, especially since many of this study's patients had multiple failed prior regimens.

Author Response

Ms. Milica Stankovic

Assistant Editor

Manuscript ID: ijms-2736106

Title: " OncoTherad® (MRB-CFI-1) Nanoimmunotherapy: a Promising Strategy to Treat Bacillus Calmette-Guérin-Unresponsive Non-Muscle-Invasive Bladder Cancer: Crosstalk Among T-cell CX3CR1, Immune Checkpoints and Toll-like Receptor 4 Signaling Pathway"

Dear Ms. Milica Stankovic:

Thank you for the e-mail dated December 06, 2023. The reviewers’ comments were of fundamental importance for the data interpretation.

In attention to the recommendations, enclosed can be found a revised manuscript, which has all the corrections indicated by the reviewer. The corrections in the manuscript are highlighted in red. A numbered list indicating how I have dealt with each of the points raised by the reviewers is also provided.

Thank you very much in advance.

Looking forward to reading from you.

Yours sincerely

Prof. Dr. Wagner José Fávaro

Laboratory of Urogenital Carcinogenesis and Immunotherapy (LCURGIM), Department of Structural and Functional Biology, Institute of Biology, University of Campinas (UNICAMP), CP-6109, 13083-865, Campinas, DP, Brazil. Telephone: + 55 (19) 3521-6104. E-mail: [email protected]

In attention to the Reviewer #2

We appreciate the reviewer's considerations, which were fundamental in enhancing the interpretation of the data.

All manuscript corrections are highlighted in red.

Comments 1: Introduction: Authors should consider elaborating on specific limitations of current BCG treatments to further underscore the need for new therapies.

Response 1: The limitations of BCG treatment, particularly therapeutic failure, as well as the consequences of radical cystectomy, have been incorporated into the Introduction. These changes are highlighted in red in the text.

Comments 2: Methods: Authors should consider moving Methods section to before Results section for readability and flow for the reader within the scientific framework.

Response 2: We agree with this reviewer's consideration. Nevertheless, we have followed to the journal's guidelines, which require the Results section preceding the Methods section.

Comments 3: Methods: Provide more details on patient selection criteria to enhance clarity of study population further. Additionally, the authors should consider a more detailed demographics table and expand on the demographic and clinical variable makeup of the patient population beyond the current parameters (age, sex, race, among others).

Response 3: It was included in the text (see 4.2 and 4.4 sections). The demographic data (age, sex, race, among others) were included in the Table 1.

Comments 4: Methods: Authors should consider including a flowchart of inclusion/exclusion criteria and patient selection.

Response 4: The flowchart of design study was included in the revised manuscript version (see Figure 8).  

Comments 5: Discussion: There should be a section within the discussion that provides an overview of any potential limitations of the study (sample size, methodology, limitations of the therapy itself, etc.) as well as clarified future directions for research and clinical applications.

Response 5: The limitations and future directions of this study were included in the Discussion section.

Comments 6: Intro and Discussion should include history of BCG and an overview of prior therapies to set the background for this novel therapy, especially since many of this study's patients had multiple failed prior regimens.

Response 6: It was included in the revised manuscript version.